# Impact of the COVID-19 Pandemic on the Acceptance and Use of an E-Learning Platform

**DOI:** 10.3390/ijerph182111372

**Published:** 2021-10-29

**Authors:** Markus Kipp

**Affiliations:** Institute of Anatomy, Rostock University Medical Center, 18057 Rostock, Germany; markus.kipp@med.uni-rostock.de

**Keywords:** education, e-learning, medicine, lockdown, COVID-19

## Abstract

E-learning has become an increasingly important part of higher education and is currently used both for distance education and as a complement to teaching on the campus. In this study, we investigated the acceptance of the e-learning platform (ELP) ClinicalKey Student (ELSEVIER©) among first-year medical students. Furthermore, we asked whether acceptance (i.e., digital activities) and user behavior changed during the COVID-19 lockdown. Two first-year medical student study cohorts were followed: one cohort during the COVID-19 lockdown and another cohort one year before the lockdown. Different parameters, such as online versus offline studying, daily activities or users versus nonusers, were recorded and evaluated. Additional surveys were conducted to understand why the students used the ELP. In the non-pandemic cohort, 68 out of 251 enrolled students registered in the ELP, while the number of registered students significantly increased during the COVID-19 lockdown (201 out of 255 enrolled 2nd semester students). The increase in registered users was paralleled by an increase in daily activities normalized per user and day. Despite this increase in ELP activities, the relative distribution of different user types (i.e., online versus offline users) did not change. To conclude, this study demonstrates that the COVID-19 lockdown increases the receptivity of medical students to an ELP, but the way the students work with the ELP remains unchanged.

## 1. Introduction

The goal of medical educational curricula is to teach medical students theoretical and clinical state-of-the art competences, in order to adequately prepare them for their later daily clinical practice. Up to the end of the last century, academic education in general, and medical education in particular, were organized mainly as face-to-face teaching sessions, such as lectures, seminars and hands-on practical courses. In recent decades, and particularly during the Corona-pandemic, digital- and/or online based teaching tools (i.e., e-learning) became part of most medical curricula [1]. In fact, digitization is not a trend, but will fundamentally change the medical profession and medical studies. In contrast to lectures, seminars or libraries, e-learning materials, if provided online, is always available and, thus, is growing in popularity among students [2].

The implementation of digital technologies in medical curricula has started globally and reached varying levels of penetration depending on local resources and demands. The current choice of digital teaching and learning formats in medicine, among both medical educators and students, is very heterogeneous, including classical static formats such as PDF and PowerPoint slides, digital problem-based learning platforms [3], gaming [4], social communication tools, audio/video-based media [5], interactive formats, and electronic testing systems, all of which can principally enrich the learning environment of a faculty.

Currently, the provision of e-learning tools or complex blended learning scenarios depends on an online platform where students and educators can access them, also called learning management systems or e-learning platforms (ELPs). In the last decade, a variety of novel text forms (e.g., multimedia books and tweets) and mediums for presenting such texts (e.g., iPad and Kindle) have been developed and successfully implemented in our daily lives. Additionally, in the academic environment, classical textbooks are increasingly supplemented by e-books used with digital devices, such as tablets, laptops or, eventually, smartphones. Such e-books or book chapters are frequently embedded into ELPs. A recent study explored differences that might exist in comprehension when students read digital versus print texts. While there were no differences across mediums when students identified the main idea of the text, students recalled key points linked to the main idea and other relevant information better when engaged with print [6]. In addition to the study performance, the acceptance of e-books among students is critical for successful implementation.

Relatively little is known regarding to what extent a sudden and unexpected “digitalization storm” [7], forced by a pandemic, impacts on the e-learning behavior of students. Starting in the spring of 2020, the outbreak of Coronavirus disease (COVID-19) forced universities worldwide, including in Germany, to close campuses and initiate (or expand) online teaching. In parallel, libraries and bookstores were closed, and consequently, the access of students to print texts was seriously limited.

Most recent studies have investigated the impact of the COVID-19 pandemic on e-learning behaviors of both students and teachers. For example, it has been shown that satisfaction with e-learning is highest among female students and students with a history of attending online classes before COVID-19 [8], and that satisfaction is the major antecedent for predicting students’ continued intention to use e-learning [9]. Beyond, it has been demonstrated that the forced implementation of e-learning, due to the COVID-19 pandemic, increased the technological skills of medical teachers, and university medical staff found that e-learning is very helpful in improving and progressing the educational process [10]. However, whether or not the acceptance and user behavior of students working with an ELP changes due to the COVID-19 lockdown has not, so far, been investigated.

## 2. Materials and Methods

### 2.1. Theoretical Framework and Hypotheses

Numerous groups have investigated different aspects of e-learning, including its acceptance [11,12,13,14], problems and challenges for implementing e-leaning [15,16], exam-related challenges [17] or gender differences [18]. Numerous theoretical models have been developed to identify the factors leading to the actual usage of e-leaning including, initially, the Technology acceptance model (TAM) [19], extended TAM versions [20], the Unified Theory of Acceptance and Use of the Technology (UTAUT) [21] and, more recently, the General Extended Technology Acceptance Model for E-Learning (GETAMEL) [22].

Previous studies have shown that most universities worldwide are using e-learning systems to deliver their curriculum as a part of their blended learning approach. While e-learning can provide benefits for teaching and learning, a high acceptance rate is pivotal. For this reason, increasing attention has been paid to uncover which factors determine the use of e-leaning tools among students. Employing various theoretical models (i.e., TAM, UTAUT, GETAMEL, etc.), various factors have been identifed that impact on the use of e-leaning plattforms. The extent to which pandemic-forced teaching shift (into the digital space) impacts on the acceptance and user behavior of students working with an ELP is unknown. Some studies, conducted during the COVID-19 pandemic, have shown an increased acceptance and appreciation of e-learning tools among doctors in training [23,24]. We, therefore, formulate the following two hypotheses:

**Hypothesis** **1** **(H1).**
*A pandemic-forced teaching shift into the digital space does not impact on the ELP acceptance among medical students.*


**Hypothesis** **2** **(H2).**
*A pandemic-forced teaching shift into the digital space does not impact on the ELP user behavior.*


### 2.2. Study Design, Participants and Study Period

The current study was designed as a single-center online study and was carried out at the University Medical Center Rostock, Institute of Anatomy, Germany. The undergraduate curriculum at the University Medical Center Rostock consists of 12 semesters, characterized by a mix of lectures, seminars, bedside teaching, practical trainings, internships and problem-based learning. The e-learning platform (ELP) Amboss was already implemented during the period of data collection. Two study cohorts were included in the study (see Figure 1A). The first cohort included students in their 2nd semester who were enrolled in 2018 for medical undergraduate studies at the University Medical Center Rostock. The second cohort included students in their 2nd semester who were enrolled in 2019 for medical undergraduate studies at the medical center. Participation in the study was voluntary and anonymous. The first cohort was granted unlimited and free access to the e-learning platform ClinicalKey Student (ELSEVIER) from 13 May until 23 July 2019 (70 days). The second cohort was granted unlimited and free access to the e-learning platform ClinicalKey Student from the 1 April until the 30 April 2020 (28 days). During this second period, practical courses, seminars and lectures had to be performed online, and the university libraries were closed due to government-imposed physical distancing regulations during the COVID-19 pandemic (red bar in Figure 1A). Therefore, the first cohort will be called non-pandemic and the second cohort pandemic.

### 2.3. Participant Recruitment

The study cohorts were informed by different approaches. The non-pandemic study cohort (enrollment year 2018, 2nd semester) was informed by group mail, announcement during the lecture, billboard advertising and flyers. Additionally, the usage of the platform was demonstrated by an employee of Elsevier (I.S.) during the dissection course 2019 (7 July 2019). The pandemic study cohort (enrollment year 2019, 2nd semester) was informed by group mail and announcement during the online lecture. Billboard advertising and flyers were, due to closure of the university during the COVID-19 crisis, not performed.

### 2.4. Data Collection Procedure

The learning management system used was ClinicalKey Student (ELSEVIER©), and deeply integrated ADOBE© Analytics delivered detailed product performance reports. ClinicalKey Student is an interactive education platform that provides access to more than 140 accepted medical textbooks covering over 40 medical specialties among anatomy, physiology and biochemistry. The ClinicalKey Student Bookshelf app allows offline access to all available textbooks. The system was accessible for all registered students at the medical school via regular internet access and the use of an individual password. Furthermore, all students had password-protected internet access to the medical school’s online library and held an institutional, separate email account. After registration to the ELP, each access was anonymously tracked, including the access date, the number of activities per day and the items used (i.e., medical textbooks or chapters). Furthermore, ADOBE© Analytics tracked whether the students simply viewed an item online or whether an item was downloaded into the digital bookshelf. The former will be called reads, while the latter will be called downloads. Moreover, the ELP allowed the users to create presentations and export these into PowerPoint for subsequent use, and this activity was tracked as an activity count. The tracking reports were provided by ELSEVIER as raw data in xls format.

At the end of each study period, students were asked to participate in an online survey. To assess medical students’ perceptions, needs and expectations when using the ELP, a standardized questionnaire was developed. Based on exploratory interviews with a professional expert, the focused subject was structured, and questions were developed. The preliminary questionnaire was validated by M.K. to narrow down the final questions. The survey was announced via group mailing, and participation was voluntary. During the online survey, respondents had the opportunity to change their answers by using a back button until they were ready for a final submission of the survey. Their responses were first documented anonymously in the confidential database the Survey Provider (Microsoft Forms) and then transferred to a local secure server. The first survey was published on 6 July 2019, and the second survey was published on the 22 May 2020.

### 2.5. Data Evaluation Procedure

Number of participants (i.e., registered users with subsequent activity), number of reads, number of downloads, and number of exports of presentations into the PowerPoint software environment per individual user and week were manually extracted from the provided xls file. Four different user types were defined prior to the data analysis: (i) nonusers visited the platform ≤4 times during the entire study period and did not download items into the digital bookshelf. (ii) Online readers read items online ≥2× per week without downloading any items into the digital bookshelf. For the nonpandemic cohort, this equals ≥20 reads during the 10-week study period, while for the pandemic cohort, this equals ≥8 reads during the 4-week study period. (iii) Balanced digital readers read items online ≥2 per week AND downloaded ≥ 3 items into the digital bookshelf. (iv) Offline readers read items online ≤2 per week AND downloaded ≥3 items into the digital bookshelf. Of note, the ELP did not track how many times a downloaded item was viewed, nor did it track the time spent at the platform per individual view/visit.

### 2.6. Statistical Analyses

Differences between groups were statistically tested using Prism (version 8.0.2, GraphPad Software Inc., San Diego, CA, USA) with confidence intervals of 0.05. *p*-values of ≤0.05 were considered to be statistically significant. The following symbols are used to indicate the level of significance: * *p* ≤ 0.05, ** *p* ≤ 0.01, *** *p* ≤ 0.001, ns indicates “not significant.” No outliers were excluded from the analyses. The Shapiro–Wilk test was applied to test for normal data distribution. Applied statistical tests are given in the respective figure legends.

## 3. Results

In a first step, we were interested in the acceptance of the ELP among our first, nonpandemic student cohort (i.e., first-year students in human medicine or dental medicine at the University Medical Center Rostock; Germany). Three distinct activities were recorded on a daily basis: first, the number of online views, second, the number of downloads to the digital bookshelf and third, the number of exports of presentations into the PowerPoint software environment. Of the 251 enrolled 2nd semester students, 68 registered at the ELP. When we plotted the weekly activities during the 10-week study period (see Figure 1B), the number of digital activities (i.e., the number of views + downloads) was low during the first 3 weeks (on average, 15.67 ± 11.89 activities per day; mean ± standard error of the mean), peaked at week 4 (441 activities per week) and dropped until the end of the first study period to 72.67 ± 23.33 activities per week. When we plotted the daily activities at week 4 (i.e., the week with the highest daily activities; see Figure 1C), we could clearly see the highest number of activities on the day of the active platform announcement (i.e., 7 July 2019; see Section 2 of this manuscript). Of note, no exam was scheduled during this period.

As demonstrated in Table 1, during the first three weeks, all the activities were exclusively online reads. After the active intervention (i.e., live demonstration), the students in the non-pandemic cohort actively used the option to download content into the digital bookshelf and continued to do so until the end of the study period. In summary, 146 items were downloaded into the digital bookshelf during the study period. The third tool, export of presentations into the PowerPoint software environment, was not used at all by this cohort (data not shown).

A closer look at the students’ online activities revealed different user types (see also Section 2 for a definition of these user types). As shown in Table 2, 28 out of the 68 participating students (~41.2%) belonged to the nonuser group, as they visited the platform ≤4 times during the entire study period and did not download any item into the digital bookshelf. Six students (~8.8%) belonged to the online reader group, as they preferentially read items online without downloading any item to the digital bookshelf (≥2 reads per week and ≤1 download; equals 20 visits). Another 5 students (~7.5%) belonged to the balanced digital reader group using both options, reading items online and downloading items to the digital bookshelf (≥2 reads per week and ≥3 downloads). Eleven students (~16.2%) belonged to the offline readers group, studying preferentially with items downloaded to the digital bookshelf (≤2 reads per week and ≥3 downloads). Eighteen students (~26.5%) could not be assigned to any of these four groups. Of note, 47% of all students used items representing ≥3 disciplines (Figure 1D).

At the end of the non-pandemic study period, an online survey was conducted, in which 12 out of 68 students participated. Of these 12 students, 42% claimed that they had used the platform daily, 33% weekly, 8% monthly and 17% less than once per month. As demonstrated in Figure 1E, although various ELP functions were used by the students, the possibility to download items (75%) and search for specific items (i.e., book collection; 67%) received the most appeal.

Next, we were interested in whether the acceptance of the digital education platform changed during the COVID-19 pandemic. We raised the following null hypothesis: A pandemic-forced teaching shift into the digital space does not impact on the ELP acceptance among medical students. While 68 out of 251 enrolled 2nd semester students from the non-pandemic cohort voluntarily registered at the ELP, 201 out of 255 enrolled 2nd semester students did so from the pandemic cohort (see Figure 2A upper part). During their 10-week trial period, the non-pandemic cohort engaged in 778 reads and 146 downloads, which averaged 77.8 reads and 14.6 downloads per week (shown in blue in Figure 2A, middle and lower part). Normalized to the study period (i.e., 70 days) and the number of users (i.e., 68), each user performed 11.4 reads and 2.1 downloads during the entire study period (i.e., 0.16 reads per user a day, and 0.03 downloads per user a day). The pandemic cohort showed, during their 4-week trial period, a total of 1559 reads and 396 downloads, which averaged 389 reads and 99 downloads per week (shown in red in Figure 2A, middle and lower part). Normalized to the study period (i.e., 28 days) and the number of users (i.e., 201), each user performed 7.8 reads and 2.0 downloads during the entire study period (i.e., 0.28 reads per user a day, and 0.070 downloads per user a day). Again, the third feature traced, export of presentations into the PowerPoint software environment, was not used at all by this cohort. To conclude, reads and downloads per user and day approximately doubled during the pandemic study period with a significant difference for the reads per user and day (*p* = 0.0059; Mann–Whitney test, see Figure 2B).

Furthermore, we raised the following second null hypothesis: A pandemic-forced teaching shift into the digital space does not impact on the ELP user behavior. As shown in Table 3, in the pandemic cohort, 67 (~33.3%) students belonged to the nonuser group, as they visited the platform ≤4 times and did not download content to the digital bookshelf. Sixteen students (~7.9%) belonged to the online reader group, as they preferentially visited online content without downloading items to the digital bookshelf. Twenty-seven students (~13.4%) belonged to the balanced digital reader group using both online content and content downloaded to the digital bookshelf. Seventeen students (~8.4%) belonged to the offline reader group, using preferential content downloaded to the digital bookshelf (≤1 visit per week and ≥3 downloads). Seventy-four students (~36.8%) could not be assigned to any of these groups. Of note, there was no obvious difference in the relative distribution of the different user types between the non-pandemic and pandemic study cohort (see Figure 2C). Although no appropriate statistic test was applied to formally test for this second hypothesis due to single observations, there rate of non-users was just slightly lower in the pandemic cohort, as was the rate of offline users. The rate of online users was unchanged, and the rate of balanced users slightly increased in the pandemic compared to the non-pandemic cohort.

At the end of the pandemic study period, an online survey was conducted, in which 109 out of 201 students participated. Of these, 33% of the students claimed that they had used the platform daily, 31% weekly, 7% monthly and 38% less than once per month, respectively. Again, we asked which ELP function was most appreciated and found that access to a broad spectrum of medical textbooks (i.e., book collection; 72%) and the possibility to download items to the bookshelf (53%) were of most appeal (see Figure 2D).

A relatively high number of students did not register at the platform or belonged to the nonuser group. During the non-pandemic period, 68 out of 251 students registered (i.e., 183 did not), and of those, 28 (41.2%) belonged to the nonuser group. In summary, 183 + 28 = 211 out of 251 (84%) did not use the provided ELP. The situation was different during the COVID-19 lockdown. During the lockdown period, 201 out of 255 students registered (i.e., 54 did not), and of those, 67 (33.3%) belonged to the nonuser group. In summary, 54 + 67 = 121 out of 255 (47%) did not use the provided ELP.

## 4. Discussion

The advancing digitization of the healthcare system requires that in the future, digital skills become an integral part of the medical curriculum. Haag and colleagues proposed that digital teaching and learning technologies should be used wherever they offer real benefits over other training scenarios [25]. To meet these challenges, systematic studies addressing the acceptance of digital media among medical students (and teachers) must be conducted.

In this study, we have tested the main hypothesis that a pandemic-forced teaching shift into the digital space does not increase the ELP acceptance among medical students. Beyond, we tested the hypothesis that a pandemic-forced teaching shift into the digital space does not impact on the user behavior of students working with an ELP. The first hypothesis was rejected, as we were able to demonstrate that the number of reads and downloads per user per day in the pandemic study cohort were roughly twice the number of reads and downloads per user per day in the non-pandemic study cohort. The difference was statistically significant for the number of reads. Beyond, while 68 out of 251 (27%) enrolled 2nd semester students from the non-pandemic cohort voluntarily registered at the ELP, 201 out of 255 (79%) enrolled 2nd semester students did so from the pandemic cohort. Of note, other factors might have influenced students ELP acceptance which are independent of the pandemic situation such as a greater availability of for example tablets or wireless internet connections.

The second hypothesis, however, was verified in our study. The relative numbers of nonusers, online readers, balanced digital readers and offline readers were comparable between the two study cohorts (see Figure 2C). To conclude, the affinity for using an e-learning platform increased during the COVID-19 pandemic but user types did not change. Beyond, we demonstrated that the active announcement of an e-learning platform is necessary to promote its broad use among students and that different user types exist with respect to e-learning platform affinity and usage.

The number of learning resources available to medical students studying for a degree in medicine is growing exponentially. In addition to traditional learning resources, such as lectures and textbooks, students are increasingly using e-learning tools, such as commercially available question banks, Wikipedia and YouTube, to supplement their learning. In a recent study, in addition to the use of traditional learning tools, the majority of students reported using a variety of e-learning tools, including online teaching videos (92%) and question banks (91%) [26]. Since online resources are sometimes of questionable or even poor quality, the acceptance of ELPs, such as ClinicalKey Student, is of importance.

That an active announcement of the used e-learning platform ClinicalKey Student was necessary to promote its broad use among students is demonstrated in Figure 1B,C. During the first three weeks, 47 activities in total were recorded by Adobe Analytics. Announcement of the ELP availability by group mail, announcement during the lecture, billboard advertising and flyers was obviously not sufficient to stimulate the students to explore a novel e-learning tool. When the ELP was actively demonstrated during the curriculum, however, daily activities significantly increased, with the highest number on the day of the announcement (i.e., 7 July; see Figure 1C). Numerous studies have focused on the impact of technology acceptance factors on students’ acceptance of e-learning. For example, it has been shown that that ease of use is a key determinant of the acceptance and usage of course websites as an effective and efficient learning technology [27]. Following the technology acceptance model, when users are presented with a new technology, a number of factors influence their decision about how and when they will use the new technology, among which are perceived usefulness (i.e.; users tend to use technology if they believe it will help them perform better) and perceived ease-of-use (i.e.; the degree to which a person believes that using a particular system will require little effort) [19]. One might speculate that due to the announcement and the presentation of the ELP, perceived ease-of-use values increased, and, consequently, daily activities also increased. However, systematic studies are required to predict and explain the use of ELPs for academic education. Of note, the increased acceptance of the ELP after 7 July 2019 cannot be attributed to exams, as the exam period started several weeks later.

One dimension of learning within the cognitive framework is referred to as “style.” The “learning style” identifiers describe how individuals acquire information and how it is processed or acted upon once acquired. In this study, we defined four different user types and asked whether the relative frequency of the user types changed during the COVID-19 lockdown. Nonusers were not interested in the ELP, online readers preferred reading medical content online without downloading content for offline studies, balanced digital readers used both options, and offline readers preferred to study with downloaded content. The different user types might well be related to the availability of appropriate end devices to work with the ELP content. For example, students owning a tablet might prefer to download content and read it offline anywhere. In contrast, students not owning a tablet might belong to the online reader group. Future studies should have to show whether this assumption is true or not.

To our surprise, a relatively high number of students did not register with the platform or belonged to the nonuser group. Of note, we do not know why a significant number of students did not use the provided ELP. Because laptops and desktops are relatively inexpensive, computer ownership by students is the norm rather than the exception. We thus consider it unlikely that non-availability of user devices is the main underlying reason. One main function of the ELP used in this study is the availability of different medical textbooks in a digital format. There are several experimental studies that have compared reading on paper versus reading on a screen and found a difference in readers’ performance, with better reading performance, when reading from paper compared to screens [28,29]. It might, thus, be possible that the non-users preferred reading printed textbooks and, therefore, did not use the ELP. Alternatively, printed textbooks were available to those students without any need to use the ELP as an alternative textbook source.

Currently, medical degree programs are increasingly focusing on self-directed and problem-based learning. This requires students to search for online resources that are of high quality and easy to retrieve. Indeed, information available on the Internet is currently being used as reference guides for sensitive health issues by nonprofessionals, physicians, and medical students. For example, in a study, students valued lecture notes (73.7%) and Wikipedia (74%) as their most important online sources for knowledge acquisition [1]. Another attractive platform with growing popularity among students from diverse disciplines is YouTube. This and similar platforms allow viewings of video content on handheld mobile devices and are not limited by time or place, unlike books, lectures, and tutorials. Although online material can principally support efficient learning, questionable information sources are sometimes used, providing incomplete or even wrong information [30,31]. Of note, it might well be that non-users preferred to work with these and other online resources rather than using the provided ELP. Medical educators should be aware of the potential influence public databases may have on students’ learning outcome, and students should be selective when using public databases, as such searches can prove challenging and time consuming, and the information gathered may be misleading due to the absence of content review. The implementation of ELPs with accepted digitalized textbooks might help medical educators determine which digital media are consumed by students.

## 5. Conclusions

This study has some limitations. First, the perceived usefulness and the perceived ease-of-use was not asked in the questionnaire and, thus, cannot be related to the different user types. Also, ADOBE© Analytics did not record on an individual user level what kind of devices were available to the students to work (or not) with the ELP, and the second hypothesis was not statistically evaluated but rather followed by an observational approach (i.e., how many students followed a specific, predefined user type).

Historically, many learning methods have been used, but in recent years, e-learning has been increasingly integrated into medical education with the expansion and dissemination of digital platforms for everyday use. However, at present, the use of digital media is not yet an integral and comprehensive component of the teaching framework of medical studies in Germany but is rather used in the sense of punctual teaching enrichment. In that context it should be pointed out that digitization and digitalization are not the same. Digitization essentially refers to taking analog data and encoding these into zeroes and ones so that computers can store, process, and transmit such information. Digitalization, in contrast, refers to the use of digital technologies with the aim to integrate new technologies into teaching and learning to improve its quality. To reach this aim, the technologies have to be used by the students.

This study demonstrates that during pandemic-related lockdowns, the acceptance of ELPs significantly increases without influencing the principal use behavior of the students. Of note, the increased use of ELPs might facilitate the observed gradual shift from paper-based reading to reading on digital devices, such as computers, tablets, or cell-phones [29]. Although this topic is rather complex, a number of studies have demonstrated that paper-based reading has several advantages over reading on digital devices, including higher reading accuracy in young children [32] and better performance in mathematics tests [33]. Although it is out of the scope of this article to discuss the entire complexity of this topic [28,29], paper-based reading clearly has several advantages which should be carefully considered when providing or integrating ELPs into medical curriculum.

## Figures and Tables

**Figure 1 ijerph-18-11372-f001:**
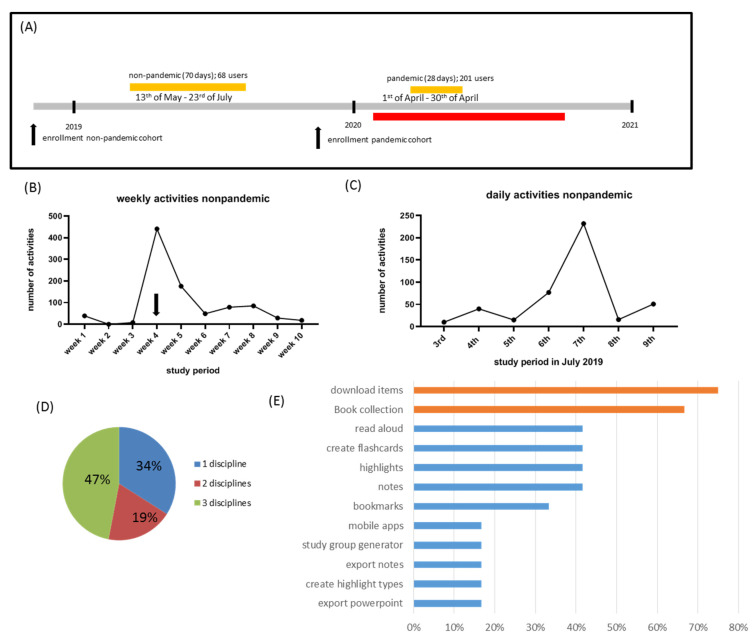
E-learning platform acceptance during the non-pandemic study period. (**A**) demonstrates the basic organization of the study. Yellow bars highlight the non-pandemic (70 days) and pandemic (28 days) study periods. The red bar highlights the period of the COVID-19 lockdown. (**B**) Weekly activities during the non-pandemic study period. The arrow highlights peak activities at week 4, which are demonstrated in (**C**) on a daily basis (i.e., from 3 until 9 July 2019). (**D**) demonstrates the percentage of users using items from one, two or three different medical disciplines. (**E**) shows survey results of the non-pandemic cohort regarding the acceptance of different e-learning platform functions among the participating students.

**Figure 2 ijerph-18-11372-f002:**
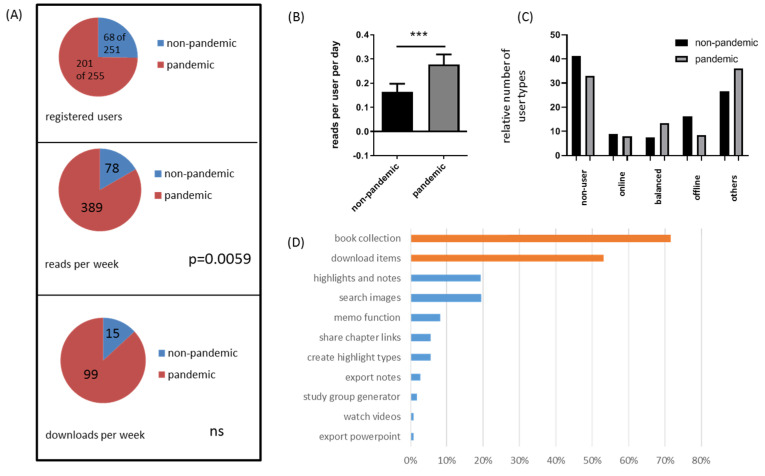
Comparative e-learning platform acceptance during the non-pandemic and pandemic study periods. (**A**) The upper part demonstrates the proportion of registered e-learning platform users during the non-pandemic (blue) and pandemic (red) study periods. The middle part demonstrates online reads per week during the non-pandemic (blue) and pandemic (red) study periods. The lower part demonstrates downloads per week during the non-pandemic (blue) and pandemic (red) study periods. Note that due to the different durations of the two study periods (see Figure 1), the results are shown as reads or downloads per week. (**B**) shows the quantitative comparison between number of reads per user per day in the pandemic and non-pandemic study cohort. Graphs show the mean and standard error of the mean (SEM). Differences between the two groups were statistically compared using the Mann Whitney test. *** equals *p* < 0.001 (**C**) shows the distribution of the different user types during the non-pandemic (black) and pandemic (gray) study periods. (**D**) shows survey results of the pandemic cohort regarding the acceptance of different e-learning platform functions among the participating students.

**Table 1 ijerph-18-11372-t001:** Number of reads and downloads during the non-pandemic study period. The number of reads and downloads during the 10-week non-pandemic study period is shown on a weekly basis. The last column shows the read/download relation.

	Reads	Downloads	Sum	Relation
week 1	39	0	39	39.0
week 2	0	0	0	0
week 3	8	0	8	8.0
week 4	380	61	441	6.2
week 5	146	30	176	4.8
week 6	37	12	49	3.1
week 7	65	14	79	4.6
week 8	68	17	85	4.0
week 9	20	9	29	2.2
week 10	15	3	18	5.0
Sum	778	146		

**Table 2 ijerph-18-11372-t002:** Number of reads and downloads of the different user types during the non-pandemic study period. The number of reads and downloads during the 10-week non-pandemic study period is shown per individual user type. The definition of a user type is summarized above the respective table. Eighteen students could not be assigned to any of these four groups and are, thus, not listed in the table.

Student ID	Reads	Downloads
non-user group (≤4 views)		
#1	1	0
#2	1	0
#3	1	0
#4	1	0
#5	1	0
#6	1	0
#7	1	0
#8	1	0
#9	1	0
#10	1	0
#11	1	0
#12	2	0
#13	2	0
#14	2	0
#15	2	0
#16	2	0
#17	2	0
#18	2	0
#19	3	0
#20	3	0
#21	3	0
#22	3	0
#23	3	0
#24	3	0
#25	3	0
#26	4	0
#27	4	0
#28	4	0
online-reader group (≥ 20 reads in total and ≤1 download)		
#1	48	1
#2	47	1
#3	45	0
#4	36	1
#5	32	0
#6	20	0
balanced digital reader (≥20 reads in total and ≥3 downloads)		
#1	130	3
#2	61	4
#3	35	8
#4	27	17
#5	27	17
offline reader group (≤20 reads in total and ≥3 downloads)		
#1	19	15
#2	18	10
#3	17	9
#4	17	10
#5	12	3
#6	11	10
#7	10	6
#8	9	7
#9	8	3
#10	5	3
#11	5	3

**Table 3 ijerph-18-11372-t003:** Number of reads and downloads of the different user types during the pandemic study period. The number of reads and downloads during the 4-week pandemic study period is shown per individual user type. The definition of a user type is summarized above the respective table.

Number	Student	Reads	Downloads
nonuser group (≤4 views)			
#1	134	4	0
#2	136	4	0
#3	150	4	0
#4	160	4	0
#5	161	4	0
#6	169	4	0
#7	190	4	0
#8	2	3	0
#9	8	3	0
#10	15	3	0
#11	19	3	0
#12	28	3	0
#13	64	3	0
#14	78	3	0
#15	115	3	0
#16	140	3	0
#17	148	3	0
#18	171	3	0
#19	196	3	0
#20	200	3	0
#21	1	2	0
#22	7	2	0
#23	18	2	0
#24	23	2	0
#25	27	2	0
#26	41	2	0
#27	53	2	0
#28	80	2	0
#29	116	2	0
#30	127	2	0
#31	139	2	0
#32	142	2	0
#33	158	2	0
#34	164	2	0
#35	165	2	0
#36	172	2	0
#37	12	1	0
#38	16	1	0
#39	42	1	0
#40	45	1	0
#41	46	1	0
#42	48	1	0
#43	70	1	0
#44	77	1	0
#45	79	1	0
#46	96	1	0
#47	107	1	0
#48	112	1	0
#49	122	1	0
#50	125	1	0
#51	128	1	0
#52	131	1	0
#53	137	1	0
#54	141	1	0
#55	147	1	0
#56	157	1	0
#57	163	1	0
#58	166	1	0
#59	182	1	0
#60	186	1	0
#61	187	1	0
#62	188	1	0
#63	189	1	0
#64	192	1	0
#65	194	1	0
#66	195	1	0
#67	198	1	0
online reader group (≥8 reads in total and ≤1 download)			
#1	37	70	1
#2	14	32	1
#3	10	24	1
#4	6	15	0
#5	17	15	1
#6	154	15	0
#7	162	14	1
#8	33	13	0
#9	38	12	1
#10	60	11	1
#11	91	10	1
#12	151	10	0
#13	167	10	1
#14	201	10	0
#15	22	8	0
#16	25	8	0
balanced digital reader (≥8 reads in total and ≥3 downloads)			
#1	43	191	55
#2	61	63	23
#3	144	50	7
#4	71	44	17
#5	199	42	3
#6	126	39	9
#7	97	36	4
#8	133	31	12
#9	3	23	5
#10	111	22	8
#11	73	21	3
#12	74	20	7
#13	88	19	3
#14	32	18	3
#15	103	18	7
#16	68	17	4
#17	168	16	3
#18	83	14	5
#19	110	14	8
#20	155	13	5
#21	58	12	7
#22	59	12	10
#23	102	11	8
#24	181	11	9
#25	11	10	3
#26	49	9	5
#27	55	8	7
offline reader group (≤8 reads in total and ≥3 downloads)			
#1	185	7	3
#2	94	6	3
#3	130	6	5
#4	170	6	4
#5	36	5	3
#6	124	5	3
#7	132	5	4
#8	175	5	3
#9	56	4	5
#10	10	4	3
#11	183	4	4
#12	106	3	3
#13	109	3	3
#14	173	3	3
#15	193	3	3
#16	47	2	5
#17	50	2	3

## Data Availability

Original data are available from the authors upon request.

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
