# Peer review of "Impact of the COVID-19 Pandemic on the Acceptance and Use of an E-Learning Platform"

_ijerph, 2021, doi:10.3390/ijerph182111372_

Round 1

Reviewer 1 Report

Minor language editing required. Authors have adequately attended to all the major issues raised previously and the article can be published. Just a note that hypotheses are normally stated in null terms and it is the null hypothesis that is tested, not the alternative.

Author Response

Thank you for the positive response and for reviewing our manuscript. We have modified the hypothesis as suggested.

Reviewer 2 Report

Fine to public.

Author Response

Thank you for the positive response and for reviewing our manuscript

Reviewer 3 Report

Thank you for revising the manuscript. I noticed that you have provided additional content and clarifications in the theoretical section of the paper. 

In terms of statistical analysis, it is not so clear how you test the first hypothesis - information about Prism has been given in general terms at some point. I suggest mentioning again the first hypothesis when you present the results of the analysis. 

The same for the second hypothesis which couldn't be tested. We found that only in the discussion. Concerning the second hypothesis, I suggest being more specific in the results section. 

Lines 366 - 382: the figures should be moved to the data section. In the discussion section, you should elaborate on the differences between using and nonusing ELP. 

Author Response

Thank you for the positive response and for reviewing our manuscript. We adopted the revised version of the manuscript following your kind suggestions.

This manuscript is a resubmission of an earlier submission. The following is a list of the peer review reports and author responses from that submission.

Round 1

Reviewer 1 Report

  1. There are no research questions, objectives or purpose of the study stated.
  2. There is no theoretical framework discussed to guide the inquiry even though the technology acceptance model is implied.
  3. Data collection strategies (instruments) are not informed by theory. e.g. aspects of TAM mentioned in the discussion but not used to design the data collection instruments.
  4. Discussion of results uses the literature as a starting point instead of the findings of the study. Reads more like a literature review of issues unrelated to the findings. 
  5. Only descriptive statistics used from which significance of results cannot be established. 
  6. Very small sample sizes used especially for the non-pandemic cohort (only 12 out of 68 users)
  7. Not clearly explained early enough what alternative platforms students used. This might suggest that students simply disliked using the ELP because they had easier to use and more accessible platforms. so conclusions about acceptance not valid. 
  8. Please see track changes in attached pdf copy for precise examples of areas of improvement.  

Author Response

Reviewer #1:

Q1:      There are no research questions, objectives or purpose of the study stated.
A1: Thank you for this comment, we now clearly state the objectives of this study. In particular we state: Hypothesis 1 (H1): A pandemic-forced teaching shift into the digital space increases the ELP acceptance among medical students.

Hypothesis 2 (H2): A pandemic-forced teaching shift into the digital space impacts on the ELP user behavior.

Q2:      There is no theoretical framework discussed to guide the inquiry even though the technology acceptance model is implied.

A2:      We agree with the reviewer’s comment and have tried to embed our study into a theoretical framework. In particular we state: Numerous groups have investigated different aspects of e-learning including its acceptance [11-14], problems and challenges for implementing e-leaning [15, 16], exam-related challenges [17] or gender differences [18]. Numerous theoretical models have been developed to identify the factors leading to the actual usage of e-leaning including, initially, the Technology acceptance model (TAM) [19], extended TAM versions [20], the Unified Theory of Acceptance and Use of the Technology (UTAUT) [21] and, more recently, the General Extended Technology Acceptance Model for E-Learning (GETAMEL) [22].

Previous studies have shown that most universities worldwide are using e-learning systems to deliver their curriculum as a part of their blended learning approach. While e-learning can provide benefits for teaching and learning, a high acceptance rate is pivotal. For this reason, increasing attention has been paid to uncover which factors determine the use of e-leaning tools among students. Employing various theoretical models (i.e., TAM, UTAUT, GETAMEL, etc.), various factors have been identifed that impact on the use of e-leaning plattforms. To what extent a pandemic-forced teaching shift into the digital space imapcts on the acceptance and user behavior of students working with an ELP is unknown. Some studies, conducted during the COVID-19 pandemic showed an increase acceptance and appreciation of e-learning tools among doctors in training [23, 24].

Q 3:     Data collection strategies (instruments) are not informed by theory. e.g. aspects of TAM mentioned in the discussion but not used to design the data collection instruments.

A3:      Thank you for this comment, the methods section was adopted accordingly.

Q 4:     Discussion of results uses the literature as a starting point instead of the findings of the study. Reads more like a literature review of issues unrelated to the findings.

A4:      We agree with the reviewer’s comment and have substantially revised the discussion accordingly. We, now, try to relate our findings to other studies and discuss our data in this context. Beyond, we state the following: In this study, we have tested the main hypothesis that a pandemic-forced teaching shift into the digital space increases the ELP acceptance among medical students. Be-yond, we tested the hypothesis that a pandemic-forced teaching shift into the digital space impacts on the user behavior of students working with an ELP. The first hypoth-esis was verified, as we were able to demonstrate that the number of reads and down-loads per user per day in the pandemic study cohort were roughly twice the number of reads and downloads per user per day in the non-pandemic study cohort. The differ-ence was statistically significant for the number of reads. Beyond, while 68 out of 251 (27%) enrolled 2nd semester students from the non-pandemic cohort voluntarily regis-tered at the ELP, 201 out of 255 (79%) enrolled 2nd semester students did so from the pandemic cohort. The second hypothesis, however, could not be verified in our study. The relative numbers of nonusers, online readers, balanced digital readers and offline readers were comparable between the two study cohorts (see figure 2C). To conclude, the affinity for using an e-learning platform increased during the COVID-19 pandemic but user types did not change. Beyond, we demonstrated that the active announcement of an e-learning platform is necessary to promote its broad use among students and that different user types exist with respect to e-learning platform affinity and usage.

Q 5:     Only descriptive statistics used from which significance of results cannot be established.

A5:      Thank you for this comment. We have statistically compared to number of reads and downloads per user number and day between the non-pandemic and pandemic study cohort. The methods section, the figures and the results section were adopted accordingly.

Q 6:     Very small sample sizes used especially for the non-pandemic cohort (only 12 out of 68 users)

A6:      In the non-pandemic cohort, of the 251 enrolled 2nd semester students, 68 registered at the ELP. From these, just 12 students did participate at the final survey. We agree that this is a very smal sample size, however, our study design did not allow to actively contact the student to participiate at the online survey. We clearly state this limitation in the revised dicsussion section of this manuscript. In particular we state: This study has some limitations. First, the perceived usefulness and the perceived ease-of-use was not asked in the questionnaire and, thus, cannot be related to the dif-ferent user types. Beyond, ADOBE© Analytics did not record on an individual user level, what kind of devices were available to the students to work (or not) with the ELP. Finally, the sample size of this study is relatively small, especially regarding the online survey participation.  

Q 7:     Not clearly explained early enough what alternative platforms students used. This might suggest that students simply disliked using the ELP because they had easier to use and more accessible platforms. So conclusions about acceptance not valid.

A7:      We agree with the reviewer’s comment and have included this part in the material and methods section. In particular we state: The e-learning platform (ELP) Amboss was already implemented during the period of data collection. We are not aware of any other ELP provided by the university to the participants. However, we cannot exclude that our student have used other ELPs or online resources, and this is now included in the discussion.

Q 8:     Please see track changes in attached pdf copy for precise examples of areas of improvement. 

A8:      Thank you for this detailed response. We have revised the manuscript accordingly.

Reviewer 2 Report

The main problem with this manuscript is that it does not reveal the authors' epistemological interest. The research questions do not provide meaningful information on this. The authors should fundamentally revise their introduction and research questions regarding their epistemological interest.

Author Response

Reviewer #2:

Q1:      The main problem with this manuscript is that it does not reveal the authors' epistemological interest. The research questions do not provide meaningful information on this. The authors should fundamentally revise their introduction and research questions regarding their epistemological interest.

A2:      Thank you very much for reviewing this paper and the positive evaluation. We now state the epistemological interest of the study in the revised version of the manuscript. Beyond, we have substantially revised the entire manuscript and hope that the overall quality was improved. We refer to the detailed comments provided to the first and third reviewer.

Reviewer 3 Report

This manuscript focuses on the use and acceptance of digital tools in medical education during COVID-19 pandemic. 

Even if the authors aimed to test the impact of the COVID-19 pandemic on the acceptance and use of an e-learning platform, I see the manuscript as a quantitative evaluation of the use of internet platforms for studying. 

Moreover, the literature and the theoretical support does not cover the complexity of the topic. 

In terms of method, it is not so clear to me the instruments for data collection. Details about the standardized questionnaire are missing. The acceptance of the digital education platform is measured as the number of visits of the platform, reads,  downloads etc. 

Besides digitization (the pure analog-to-digital conversion of existing data and documents), I encourage the authors to discuss also digitalization (“the use of digital technologies to change a business model and provide new revenue and value-producing opportunities; it is the process of moving to a digital business)”. 

Author Response

Reviewer #3:

Q1:      Even if the authors aimed to test the impact of the COVID-19 pandemic on the acceptance and use of an e-learning platform, I see the manuscript as a quantitative evaluation of the use of internet platforms for studying.

A1:      Thank you for this comment, we now clearly state the aims and hypothesis of the study. Beyond, we have modified the entire manuscript including the abstract using more precise terms such as digital activities or ELP activities. In particular we state: Hypothesis 1 (H1): A pandemic-forced teaching shift into the digital space increases the ELP acceptance among medical students.

Hypothesis 2 (H2): A pandemic-forced teaching shift into the digital space impacts on the ELP user behavior.

Q2:      Moreover, the literature and the theoretical support does not cover the complexity of the topic.

A2:      Thank you for this comment, the manuscript was substantially modified and we hope that the complexity of study is now supported.

Q3:      In terms of method, it is not so clear to me the instruments for data collection. Details about the standardized questionnaire are missing. The acceptance of the digital education platform is measured as the number of visits of the platform, reads, downloads etc.

A3:      Thank you for this comment. We have adopted the manuscript accordingly and hope that these points are clearer now.  First, we state what items were included in the online survey. Second, we have modified the term “acceptance” to clearly define the term. Third, we cannot say what kind of device our students have used during their work with the ELP. However, we now clearly state how the data were collected. In particular we state: The learning management system used was ClinicalKey Student (ELSEVIER©), and deeply integrated ADOBE© Analytics delivered detailed product performance re-ports. ClinicalKey Student is an interactive education platform that provides access to more than 140 accepted medical textbooks covering over 40 medical specialties among anatomy, physiology and biochemistry. The ClinicalKey Student Bookshelf app allows offline access to all available textbooks. The system was accessible for all registered students of the medical school via regular internet access and the use of an individual password. Furthermore, all students had password-protected internet access to the medical school´s online library and held an institutional, separate email account. After registration to the ELP, each access was anonymously tracked, including the access date, the number of activities per day and the items used (i.e.; medical textbooks or chapters). Furthermore, ADOBE© Analytics tracked whether the students simply viewed an item online or whether an item was downloaded into the digital bookshelf. The former will be called reads, while the latter will be called downloads. Moreover, the ELP allowed the users to create presentations and export these into PowerPoint for subsequent use, and this activity was tracked as an activity count. The tracking reports were provided by ELSEVIER as raw data in xls format.

Q4:      Besides digitization (the pure analog-to-digital conversion of existing data and documents), I encourage the authors to discuss also digitalization (“the use of digital technologies to change a business model and provide new revenue and value-producing opportunities; it is the process of moving to a digital business)”.

A4:      Thank you for this comment, we have adopted the last part of the discussion section accordingly. In particular we state: This study has some limitations. First, the perceived usefulness and the perceived ease-of-use was not asked in the questionnaire and, thus, cannot be related to the different user types. Beyond, ADOBE© Analytics did not record on an individual user level, what kind of devices were available to the students to work (or not) with the ELP. Finally, the sample size of this study is relatively small, especially regarding the online survey participation.  

Historically, many learning methods have been used, but in recent years, e-learning has been increasingly integrated into medical education with the expansion and dissemination of digital platforms for everyday use. However, at present, the use of digital media is not yet an integral and comprehensive component of the teaching framework of medical studies in Germany but is rather used in the sense of punctual teaching enrichment. In that context it should be pointed out that digitization and digitalization is not the same. Digitization essentially refers to taking analog data and encoding these into zeroes and ones so that computers can store, process, and transmit such information. Digitalization, in contrast, refers to the use of digital technologies with the aim to integrate new technologies into teaching and learning to improve its quality. To reach this aim, the technologies have to be used by the students.

Round 2

Reviewer 3 Report

I applaud the authors' endeavor to clarify the issues raised by the reviewers. However, I see the manuscript as a descriptive image of the topic with little contribution to the literature. 

Actually, the paper has serious methodological problems:

The additional information on the instrument for the data collection is not sufficient. The psychometric properties of the scale and some examples of items are still missing. Overall, the way of comparison between non-pandemic and pandemic use of platform seems strange. The most valid comparison should be within a group, instead of between groups. 

Consequently, I encourage you to further enrich the theoretical foundations and explore the data in order to point some potential mechanisms and advance some explanations in this matter. 

response:

Thank you for this comment. First, the psychometric properties of the scale, if available, are clearly shown in figure 2. For some others, they simply don’t exist and, thus, cannot be given. Second, we agree the best comparison would be within a group, instead of between groups, however, the study was not designed in this way and, thus, those kind of comparisons cannot be done post-hoc.